# Deep-Learning-Based ADHD Classification Using Children’s Skeleton Data Acquired through the ADHD Screening Game

**DOI:** 10.3390/s23010246

**Published:** 2022-12-26

**Authors:** Wonjun Lee, Deokwon Lee, Sanghyub Lee, Kooksung Jun, Mun Sang Kim

**Affiliations:** School of Integrated Technology, Gwangju Institute of Science and Technology, Gwangju 61005, Republic of Korea

**Keywords:** ADHD, deep learning, screening, skeleton

## Abstract

The identification of attention deficit hyperactivity disorder (ADHD) in children, which is increasing every year worldwide, is very important for early diagnosis and treatment. However, since ADHD is not a simple disease that can be diagnosed with a simple test, doctors require a large period of time and substantial effort for accurate diagnosis and treatment. Currently, ADHD classification studies using various datasets and machine learning or deep learning algorithms are actively being conducted for the screening diagnosis of ADHD. However, there has been no study of ADHD classification using only skeleton data. It was hypothesized that the main symptoms of ADHD, such as distraction, hyperactivity, and impulsivity, could be differentiated through skeleton data. Thus, we devised a game system for the screening and diagnosis of children’s ADHD and acquired children’s skeleton data using five Azure Kinect units equipped with depth sensors, while the game was being played. The game for screening diagnosis involves a robot first travelling on a specific path, after which the child must remember the path the robot took and then follow it. The skeleton data used in this study were divided into two categories: standby data, obtained when a child waits while the robot demonstrates the path; and game data, obtained when a child plays the game. The acquired data were classified using the RNN series of GRU, RNN, and LSTM algorithms; a bidirectional layer; and a weighted cross-entropy loss function. Among these, an LSTM algorithm using a bidirectional layer and a weighted cross-entropy loss function obtained a classification accuracy of 97.82%.

## 1. Introduction

Attention deficit hyperactivity disorder (ADHD) is a disorder that occurs frequently in childhood and refers to a condition in which attention is continuously insufficient, resulting in distraction, hyperactivity, and impulsivity [1]. ADHD patients range from children to adults, but the proportion of adolescents and children among ADHD patients accounts for more than 80% of the total number of patients. In addition, the number of children and adolescents with ADHD is increasing every year [2]. If ADHD is left untreated, its symptoms can lead to difficulties throughout childhood [3], and children with ADHD symptoms have a 70% chance of developing ADHD into adolescence. Furthermore, if the condition remains untreated, it can persist into adulthood for more than 50% of children. For these reasons, early diagnosis and treatment of ADHD are important [4,5].

ADHD is a psychiatric disease, not a fracture or cancer that can be diagnosed relatively simply using MRI or CT scan results. Psychiatric diseases are not simple diseases that can be diagnosed with simple tests or symptoms, and therefore, doctors require a large amount of time and effort for diagnosis. Currently, the most commonly used method for diagnosis of ADHD by doctors in hospitals involves aggregating the results of consultations with child patients, consultations with their parents or teachers, and survey results such as the child behavior checklist (CBCL) [6,7]. In addition, the results of audio–visual attention tests such as the continuous performance test (CPT) are used as auxiliary data for the diagnosis of ADHD [8,9]. However, it is not easy to make an accurate diagnosis only on the basis of the above results because surveys and counseling, which have the greatest influence on diagnosis, involve a large number of subjective opinions of parents and the children’s teachers [10]. In addition, since it takes a large amount of time to undertake counseling and surveying, doctors require a great amount of effort to conduct ADHD diagnostic tests. There is also a global shortage of trained specialists able to diagnose ADHD, which often delays diagnosis.

As mentioned above, the existing method of diagnosing ADHD by synthesizing the results of counseling and questionnaire results is based on the activity level of children observed by parents or teachers, and thus subjective errors may occur depending on the observer. In order to reduce errors due to subjective judgment, studies are underway to reduce subjective errors by using accelerometers or simulators to objectively measure and judge the activity level of children and to make more accurate diagnoses with objective results.

While various studies as above are being conducted, research on machine learning and deep learning has accelerated, and we have started to apply machine learning and deep learning and develop algorithms for classification in various fields in various medical fields [11,12]. In particular, with the development of machine learning and deep learning technologies, studies on the diagnosis of ADHD through artificial intelligence using a variety of objective data have been actively conducted since 2010. Research is underway for faster and more efficient ADHD diagnosis through machine learning and deep learning using data acquired through accelerometers, simulators, and games, as well as biometric data such as MRI, EEG, and ECG as input data. These research results are expected to help doctors improve the accuracy of ADHD diagnosis and shorten the time taken for this process. In particular, in the case of MRI, the Neuro Bureau ADHD-200 Dataset [13] was released, and various research institutes are now actively conducting research to improve the accuracy of ADHD diagnosis using this dataset by developing machine learning and deep learning algorithms.

In this study, a game was developed for children to screen for ADHD in a child-friendly environment such as a school without direct intervention from experts. While the children were playing the game, the children’s skeleton data were acquired using five depth sensors. Using the acquired skeleton data and deep learning algorithm, ADHD, ADHD-RISK, and normal were screened with 97% accuracy. ADHD screening evaluation was conducted using several simple models, and among them, it was verified that the LSTM-bidirectional model had the highest accuracy. In particular, ADHD-RISK is a class that has not been seen in other studies and is a taxon that is difficult for doctors to discriminate in actual clinical practice. However, it is important to screen ADHD-RISK that has the potential to develop into ADHD with high accuracy in actual ADHD clinical practice, and this system is expected to be helpful in the future.

The detailed structure of this paper is as follows. The Introduction section introduces the existing methods for screening ADHD. The Materials and Methods section introduces the materials used in this study and the methods proposed. The Results section describes the experimental results. The Discussion section describes the discussion in this study. Finally, the Conclusion section presents the conclusion.

## 2. Related Work

Various data acquisition systems and artificial intelligence algorithms for the diagnosis of ADHD are being developed. As a first example, various research teams have developed deep learning or machine learning algorithms using the Neuro Bureau ADHD-200 Dataset and have conducted studies on the diagnosis of ADHD. The ADHD-200 Dataset consists of 776 resting-state fMRI and structural MRI data [14,15,16,17,18,19,20,21].

The team of Peng et al. developed a CNN-based deep learning algorithm to obtain an ADHD diagnosis accuracy of 72.9% [16]. In addition, the research team of Chen et al. developed an SVM-based machine learning algorithm and acquired an ADHD diagnosis accuracy of 88.1% [21]. Since a very good public dataset is publicly available for MRI-data-based ADHD diagnosis research, it is expected that many research teams will be able to achieve higher accuracy research results through continuous algorithm development.

The second example comprises studies on the diagnosis of ADHD using EEG data [22,23,24,25,26,27,28]. The team of Tosun et al. obtained 92.2% ADHD classification accuracy using an LSTM-based deep learning algorithm for 1088 ADHD patients and 1088 normal groups [25]. In addition, the research team of Altinkaynak et al. obtained an accuracy of 91.3% using an MLP-based machine learning algorithm using EEG data of 23 ADHD patients and 23 normal subjects [27].

A third related type of study involves the use of continuous performance test (CPT) test results; this test is widely used for the diagnosis of ADHD in hospitals. CPT test results are used as input data for ADHD classification research [29,30]. Using the CPT results of 213 ADHD patients and 245 normal subjects, the research team of Slobodin et al. obtained an ADHD classification accuracy of 87% using a random-forest-based machine learning algorithm [29].

The last related study was conducted by the research team of O’Mahony et al., who classified ADHD using data measured during the TOVA test by having the children under study wear two IMU sensors on their waist and ankle. A classification accuracy of 95.1% was obtained using the SVM-based machine learning algorithm [31].

As mentioned above, various datasets and algorithms are being used and researched for the classification of ADHD. However, studies on the classification of ADHD using skeleton data have not yet been conducted. Skeleton data comprise the subject’s joint movements. After giving children a specific task, we acquired skeleton data and RGB images while the children were performing the task. After all data were acquired, four psychiatrists divided the children into three categories on the basis of the RGB image analysis results and the CBCL and K-ARS results: ADHD, ADHD-RISK, or normal. In this study, it was assumed that there would be a significant difference in the behaviors of children with ADHD, children with ADHD-RISK, and normal children while performing tasks. Using the skeleton data measured on the basis of the above hypothesis as input data, a deep-learning-based algorithm was used to classify the ADHD, ADHD-RISK, and normal groups.

## 3. Materials and Methods

### 3.1. Description of the Game for the Screening Diagnosis of ADHD

In this study, an ADHD diagnosis game was used to acquire skeleton data for children’s ADHD screening diagnosis.

The game consists of a total of five stages, comprising two practice games and three main games. In the game, a robot first moves randomly on the nine numbered boards marked on the floor, as shown in Figure 1a. At this time, the child memorizes the path the robot takes while waiting in the wait zone, as shown in Figure 1b. After the robot has completed moving on the path, the child has to follow the path the robot has taken after the start signal given by the robot. While following the path, the child performs one more task.

As the child follows the path, the characters of the witch or the Wizard of Oz appear on the screen. As shown in Figure 2a, when a witch appears, the children sit down, and when a character appears, the children wave their hands, as shown in Figure 2b. The higher the level, the more complex the path. For more information about the game, please see Ref. [32].

### 3.2. Skeleton Capture System Using Azure Kinect

As shown in Figure 3, five Azure Kinect (Microsoft Corp., Redmond, WA, USA) units from Microsoft and a beam projector (HU85LA) from LG were used to acquire the children’s skeleton data while the children were playing the game. Moreover, Robocare’s robot (Silbot) was used to progress the game and follow the path. In this study, the human tracking and skeleton merge algorithm using Azure Kinect was adopted as in a previous study by this research team (please refer to [33] for details).

In the sensor system, a calibration procedure was performed to match the coordinate system of each sensor. For this, trajectory data consisting of 3D centroid coordinates of a sphere object with a specific color were used. The trajectory data recognized by each subordinate sensor were used to calculate the rotation matrix using the trajectory data extracted by the master sensor. In addition, the coordinate system of the entire system calibrated with the coordinate system of the master sensor was converted to the world coordinate system set by the user using the Aruco marker. After that, the skeleton data of the target in the capturing area can be extracted from the depth image using Kinect Azure Body Tracking SDK. The skeleton data extracted from each sensor are merged with the skeleton data extracted from the other sensors that recognize objects from different angles to overcome the occlusion problem that occurs in a single sensor. The merging procedure is performed for each joint and includes a method of filtering the noise candidate group using DBSCAN based on the location information of each joint. As a result, the system was able to extract the skeleton data of the object in the capturing area of 3 m × 5 m from the world coordinate system set by the user by using five sensors installed at a height of 2 m. Finally, through the above system, the movement of each waiting stage and game stage acquired continuous movement at 15 frames per second.

The beam projector was used to show the path for the main game and the witch or character corresponding to the additional task. Finally, the robot was responsible for explaining and demonstrating the game to the children.

### 3.3. Data Acquisition Methods and Target Candidates

For this study, elementary school students from first to sixth grade living in Seoul, South Korea, were recruited. Data were acquired from October 2019 to December 2021 from a total of three elementary schools and one research center. All children participating in the experiment were given parental consent and went through an ethical review process.

The children’s skeleton data and RGB image data were saved while the children played the game. The data were largely divided into standby data, which were acquired while the robot explained the game and demonstrated the path the child should follow, and game data, which were acquired while the child played the game. All the children played the game in five stages. Therefore, for each stage, two pieces of data were generated during standby and during the game, resulting in a total of 10 pieces of data. Additionally, the standby data acquired while the robot explained the whole game before the first game was included. In conclusion, each child had six standby data points and five game data points, totaling 11 data points. A total of 596 children participated in this study. By synthesizing the CBCL and K-ARS results performed before the game for this study and the RGB image viewing results among the obtained children’s data, four clinicians divided them into the ADHD, ADHD-RISK, and normal groups.

As shown in Table 1, as a result of classification, 66 ADHD group children, 181 ADHD-RISK group children, and 349 normal group children were classified.

### 3.4. Deep Learning Model Using GRU, RNN, and LSTM for ADHD Classification

The RNN-based deep learning algorithm was used to classify the ADHD, ADHD-RISK, and normal groups. The size of the data for each level was different because the standby time and the completion time of the game were different for each level. Therefore, the longest frame among all children’s step-by-step data was defined as the reference size. If each data length was smaller than the standard size, the remaining frames were set to zero. The Azure Kinect devices used in this study can provide data for a total of 32 joints. However, in this study, a total of 18 joints were used, except for the low-accuracy and unstable end point joints, as shown in Figure 4 [19].

A total of six standby skeleton data points and a total of five game data points were used as input data. As shown in Figure 5, each sequential data point passed through the RNN layer. Each feature extracted after passing through the RNN was concatenated. Finally, they passed through the classification layer and children were finally classified as ADHD, ADHD-RISK, or normal. The RNN models used were GRU, RNN, and LSTM.

Moreover, as shown in Figure 6, we experimented by adding a bidirectional layer to each model to improve performance. The final output y was obtained by concatenating the hidden states $\underset{h}{\to }$ and $\underset{h}{\leftarrow }$ of the t  cell of the forward RNN-based model and the t−1 cell of the backward RNN-based model. ReLU was used as the activation function. In addition, as shown in Equation (1), a weighted cross-entropy loss function was used among the loss functions to prevent overfitting and to improve performance because the number of data points for each class was different.
(1)$$Weighted\;Cross\;Entropy\;Loss=-\sum_{i=1}^{C}w_{i}t_{i}\log\left(p_{i}\right)$$

In formula (1), $t_{i}$ is the truth label, $p_{i}$ is the Softmax probability for the $i^{th}$ class, and $w_{i}$ represents the weight of the loss function. The weight is given as the inverse of the rate of input data. In addition, this experiment was verified by leave-one-person-out cross-validation.

## 4. Results

In this study, classification of ADHD, ADHD-RISK, and normal groups was performed using the RNN model using skeleton data as input data. Three types of RNN models were used: GRU, RNN, and LSTM, and each model used a bidirectional layer and a weighted loss function to improve performance and prevent overfitting. Moreover, in this paper, accuracy, precision, recall, and the F1-score were used to evaluate the model. The formula for each parameter is as shown in the following Formulas (2)–(5):(2)$$Accuracy=\frac{TP+TN}{TP+TN+FP+FN}$$
(3)$$Precision=\frac{TP}{TP+FP}$$
(4)$$Recall=\frac{TP}{TP+FN}$$
(5)$$F1=\frac{2\times Precision\times Recall}{Precision+Recall}$$

In the above formulas, TP (true positive) is the result of predicting an answer that is actually true as true; FP (false positive) is the result of predicting an answer that is actually false as true; FN (false negative) is the result of predicting an answer that is actually true as false; and finally, TN (true negative) is the result of predicting an answer that is actually false as false.

As can be seen in Table 2, GRU, RNN, and LSTM showed 94.04%, 88.35%, and 88.35% accuracy, respectively. However, it was confirmed that the F1-score of the ADHD and ADHD-RISK classes did not exceed 90 and was low.

Thus, in order to improve the performance, we used a bidirectional layer and a weighted loss function in the above model.

As a result, as shown in Table 3, the accuracy of each model slightly increased to 96.81%, 96.81%, and 97.82% for GRU, RNN, and LSTM, respectively. The accuracy increased slightly, but the F1-score increased significantly. Among the F1-scores of each model, the ADHD and ADHD-RISK classes scored 51~89% before using the bidirectional layer and weighted loss function but these scores increased significantly to 87~100% after using the bidirectional layer and weighted loss function. In particular, the ADHD class of the GRU-bidirectional weighted loss function recorded an F1-score of 100%.

After confirming the above results, we conducted one additional experiment. The skeleton data used in this study were divided into two main categories: the skeleton data of children waiting while the robot explained and demonstrated the game before the game started, and skeleton data obtained while the children directly played the game. We verified which of the above two large categories of data were helpful for the classification of ADHD. This test used models that applied a bidirectional layer and a weighted loss function to each RNN model on the basis of the previous experiment. Table 4 below shows the results of using only the skeleton data during standby and the game.

When only the skeleton data were used during standby, the RNN model obtained the highest accuracy with 96.14%, and when only the skeleton data were used during the game, the GRU model obtained the highest accuracy with 95.39%. The accuracy was similar for standby data and game data, but there was a significant difference in the F1-scores of ADHD and ADHD-RISK. We think that it is more important to screen ADHD-RISK or ADHD than to screen children in the normal group in the ADHD screening diagnosis. This is because, if a child with ADHD is screened as normal, an opportunity for early treatment may be missed [18]. When only waiting data were used, the F1-Score of ADHD-RISK was higher by at least 5% to maximum 14% for each model. In addition, in the case of ADHD, the F1-Score of the waiting data was 2% higher, except for the GRU model. On the basis of the above results, it was judged that the skeleton data during waiting were more helpful in classifying ADHD and ADHD-RISK. Since the validation method of the model used in this study is leave-one-out cross-validation, in order to obtain the standard deviation of the model result, it is necessary to check the results of learning several times for each model. The standard deviation of accuracy after learning five times for each model is 0.17~0.67%.

## 5. Discussion

We classified ADHD, ADHD-RISK, and normal groups using only the skeleton data of children acquired through games. Currently, there are no studies using skeleton data among studies for the screening diagnosis of children’s ADHD. All children who participated in the experiment for this study performed the same task in the same limited environment. The advantage of the children’s data collected under these conditions was that they were objective and that high-quality data could be acquired in a much shorter time than the method of acquiring data by wearing an IMU device for a long period of time. This study is expected to serve as a cornerstone for future research on the screening and diagnosis of ADHD in children using skeleton data. Since the data used in this study were time series data, they were compared using an RNN-type model, and as a result, an accuracy of 97.82% was obtained. High accuracy was verified in this study, even when compared with the results of other studies that classified ADHD using various data. The model with the highest accuracy was an LSTM model with a bidirectional layer added, and the confusion matrix obtained through the model is shown in Figure 7 below.

Figure 7 is the confusion matrix of the best results obtained using the GRU, RNN, and LSTM models with bidirectional layers added and the entire skeleton data, respectively. All models correctly selected the normal group and the ADHD group. However, differences in overall accuracy occurred due to differences in ADHD-RISK screening results. In particular, the LSTM model classified all 349 out of 349 people into the normal group, and 65 out of 66 people into the ADHD group. However, in the case of ADHD-RISK, 7 out of a total of 181 ADHD-RISK patients were classified into the normal group, and 5 people were classified into the ADHD group. Classifying the ADHD-RISK group into the ADHD group is not a major problem but classifying the ADHD-RISK group as normal is to be overcome through future research on the performance improvement of the model. The diagnosis of ADHD is actually conducted in hospitals with various indicators. ADHD is not a disease that can be objectively diagnosed through the results of MRI or CT scans such as with cancer, but instead requires a mental evaluation. For this reason, this study was established, and the deep learning algorithm used in this study and the skeleton data obtained through the game comprised a system created to selectively diagnose ADHD in order to help doctors prior to diagnosis. The goal of this study was to enable children to easily perform screening tests through games, not only in schools but also in educational centers, before receiving an accurate diagnosis by a doctor using this system. Through this system, it is possible to judge the situation of children on the basis of the children’s ADHD screening results, to recommend hospital treatment to children with ADHD, and to guide parents and teachers in charge of ADHD-RISK children to monitor those children.

This study used a restricted game environment to observe children’s behavior. During the game, the children’s skeleton data were acquired. ADHD, ADHD-RISK, and normal were classified using the RNN-based deep learning algorithm using the acquired children’s skeleton data. ADHD-RISK was used as a new class in this paper. When doctors diagnose ADHD and normal, the difference between the two taxa is clear, but ADHD-RISK, which lies on the border between ADHD and normal, is difficult to classify. In fact, by classifying ADHD-RISK, which is more needed by doctors in clinical practice, we have developed a system that can help doctors screen for ADHD in future clinical practice.

## 6. Conclusions

In this study, we classified ADHD, ADHD-RISK, and normality by using children’s skeleton data acquired through a game for the screening and diagnosis of children’s ADHD. The performance of various RNN models was compared, and among them, the LSTM-bidirectional model showed the best results. In the future, we plan to conduct research on performance improvement using various models such as the GCN model optimized for skeleton data. We also intend to analyze which joints affect the model at any point in time using the attention layer. On the basis of the above results, we would like to simplify the skeleton data acquisition system used in this study by utilizing a different number of existing sensors and proposing a new game system that can focus on the major joints in ADHD classification.

## Figures and Tables

**Figure 1 sensors-23-00246-f001:**
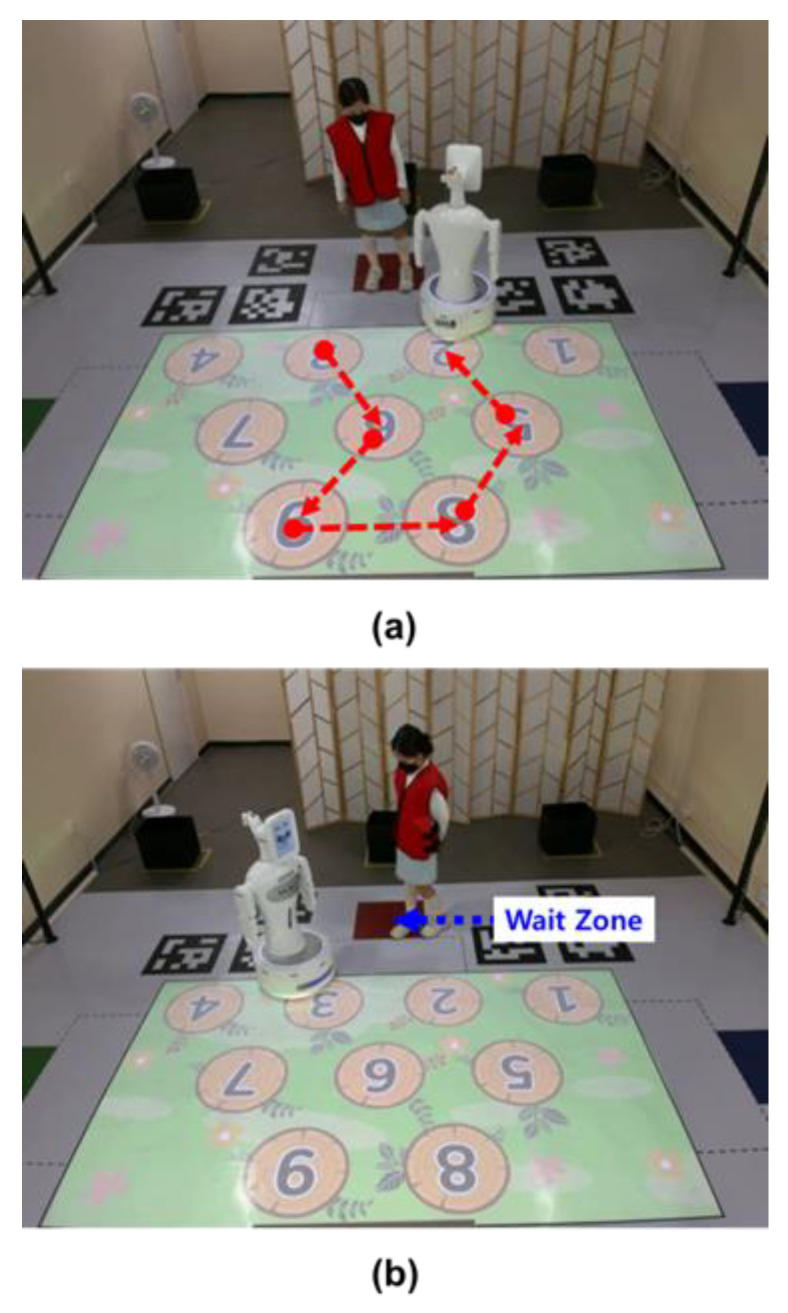
A game for the diagnosis of ADHD. (**a**) The robot moves on the numbered board first. (**b**) A child waiting in the waiting zone for the robot to take a certain path remembers the path taken by the robot and follows it.

**Figure 2 sensors-23-00246-f002:**
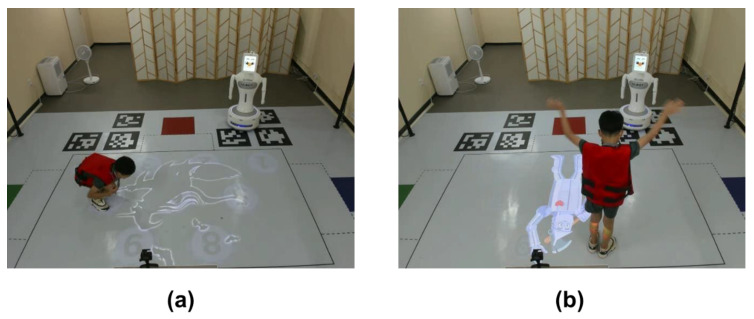
Children perform additional tasks while following the robot’s path. (**a**) When the witch appears on the floor, the child sits down. (**b**) When the character appears, the child waves both hands over their head.

**Figure 3 sensors-23-00246-f003:**
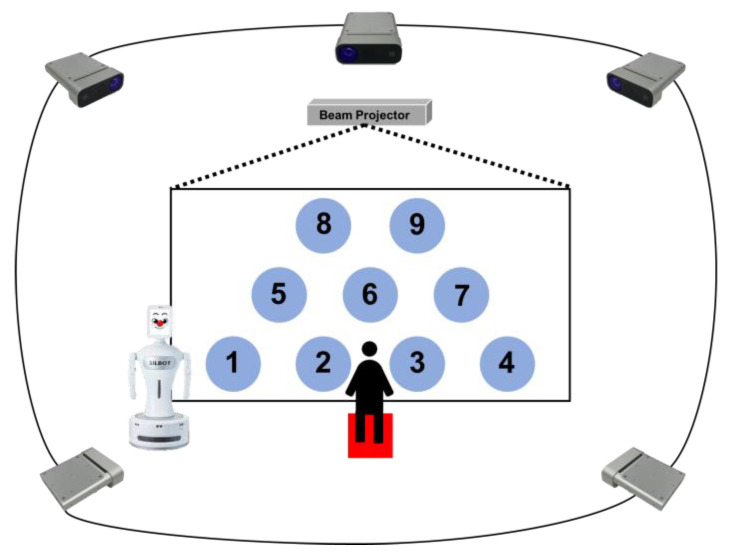
Game setup for children, consisting of five Azure Kinect units to acquire the children’s skeleton data, a beam projector to project the game screen, and a robot to help the game progress.

**Figure 4 sensors-23-00246-f004:**
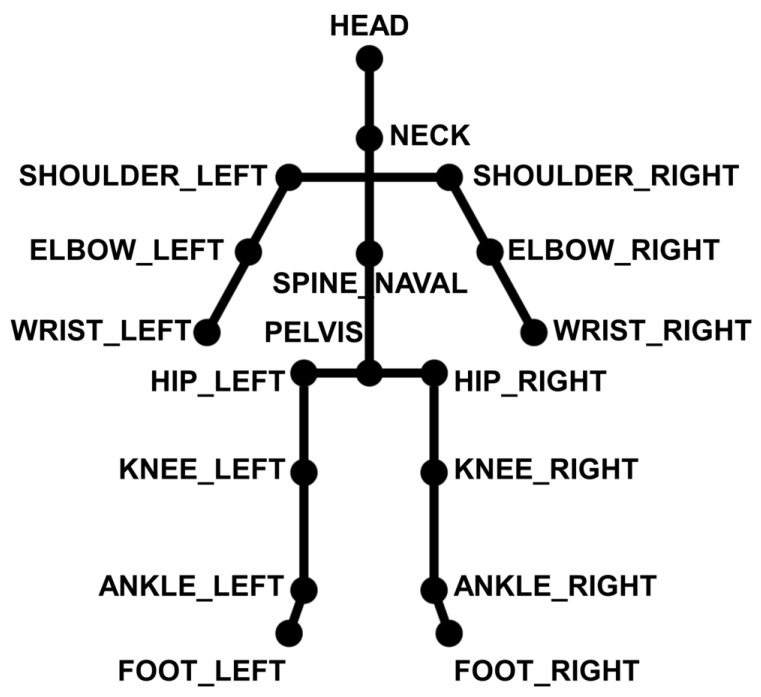
Joints used to classify ADHD classes in children.

**Figure 5 sensors-23-00246-f005:**
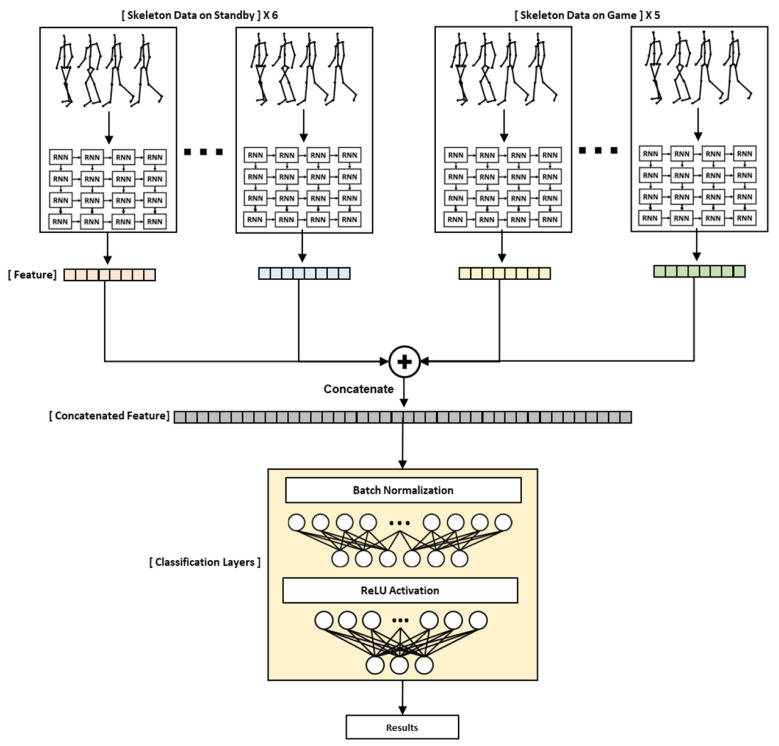
Description of an RNN-based model designed to classify children’s ADHD class.

**Figure 6 sensors-23-00246-f006:**
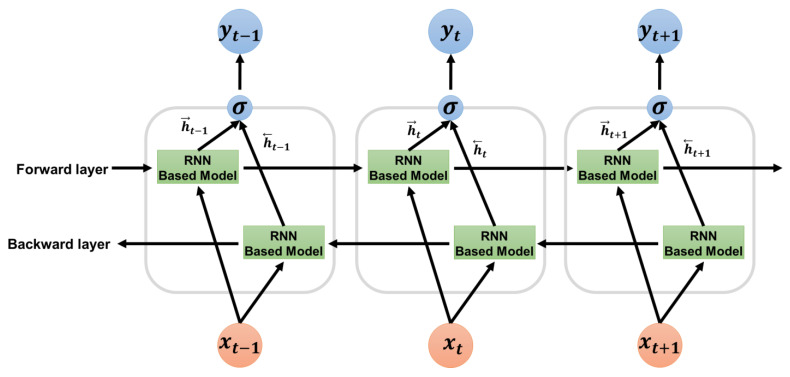
Bidirectional layer description for improving the performance of the RNN-based ADHD classification model.

**Figure 7 sensors-23-00246-f007:**
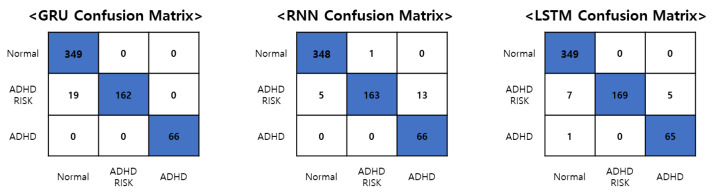
The confusion matrix of the GRU, RNN, and LSTM model with the best classification results.

**Table 1 sensors-23-00246-t001:** Number of children by ADHD class of children participating in this study.

Classes	Number of Participants
ADHD	66
ADHD-RISK	181
Normal	349
Total	596

**Table 2 sensors-23-00246-t002:** Comparison of results of three RNN-based ADHD classification models using skeleton data as input data.

GRU				
**Label**	**Accuracy**	Precision	Recall	F1-Score
Normal	94.04	0.93	1	0.96
ADHD-RISK		1	0.70	0.82
ADHD		0.96	0.82	0.89
RNN				
Label	Accuracy	Precision	Recall	F1-Score
Normal	88.35	0.87	1	0.93
ADHD-RISK		1	0.54	0.70
ADHD		0.89	0.48	0.62
LSTM				
Label	Accuracy	Precision	Recall	F1-Score
Normal	88.35	0.86	1	0.92
ADHD-RISK		1	0.63	0.77
ADHD		1	0.34	0.51

**Table 3 sensors-23-00246-t003:** Using skeleton data as input data and comparing the results of adding a bidirectional layer and a weighted loss function to three RNN-based ADHD classification models.

GRU-Bidirectional with Weighted Loss				
Label	Accuracy	Precision	Recall	F1-Score
Normal	96.81	0.95	1	0.97
ADHD-RISK		1	0.89	0.94
ADHD		1	1	1
RNN-Bidirectional with Weighted Loss				
Label	Accuracy	Precision	Recall	F1-Score
Normal	96.81	0.98	0.99	0.99
ADHD-RISK		0.99	0.90	0.94
ADHD		0.83	1	0.91
LSTM-Bidirectional with Weighted Loss				
Label	Accuracy	Precision	Recall	F1-Score
Normal	97.82	0.97	1	0.98
ADHD-RISK		1	0.93	0.96
ADHD		0.92	0.98	0.95

**Table 4 sensors-23-00246-t004:** ADHD class classification results when using the skeleton data acquired during waiting and skeleton data acquired during the game as input data.

Standby Skeleton					Game Skeleton				
GRU-Bidirectional with Weighted Loss					GRU-Bidirectional with Weighted Loss				
Label	Accuracy	Precision	Recall	F1-Score	Label	Accuracy	Precision	Recall	F1-Score
Normal	95.57	0.96	0.98	0.97	Normal	95.39	0.95	1	0.97
ADHD-RISK		0.97	0.78	0.86	ADHD-RISK		1	0.69	0.81
ADHD		0.87	0.97	0.91	ADHD		0.89	1	0.94
RNN-Bidirectional with Weighted Loss					RNN-Bidirectional with Weighted Loss				
Label	Accuracy	Precision	Recall	F1-Score	Label	Accuracy	Precision	Recall	F1-Score
Normal	96.14	0.95	0.99	0.97	Normal	95.12	0.98	1	0.99
ADHD-RISK		1	0.90	0.94	ADHD-RISK		1	0.67	0.80
ADHD		0.89	0.93	0.91	ADHD		0.81	1	0.89
LSTM-Bidirectional with Weighted Loss					LSTM-Bidirectional with Weighted Loss				
Label	Accuracy	Precision	Recall	F1-Score	Label	Accuracy	Precision	Recall	F1-Score
Normal	94.85	0.94	1	0.97	Normal	94.85	0.95	1	0.97
ADHD-RISK		0.95	0.70	0.81	ADHD-RISK		1	0.65	0.79
ADHD		0.94	0.91	0.92	ADHD		0.83	1	0.90

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
