# Peer review of "Deep-Learning-Based ADHD Classification Using Children’s Skeleton Data Acquired through the ADHD Screening Game"

_sensors, 2022, doi:10.3390/s23010246_

Round 1

Reviewer 1 Report (Previous Reviewer 1)

The research article title "Deep-Learning-Based ADHD Classification Using Children’s Skeleton Data Acquired through the ADHD Screening Game" interesting and authors have worked on it to improve for consideration in Sensors(MDPI). Certain modifications are noticed and now manuscript is significantly improved.

I want few incorporation in the Introduction Section which will help readers:

(1) Add few lines to mention the organization of the paper.

(2) Clearly mention the research contribution of this paper. It will be good if authors mention the research gaps which are filled in this paper. (preferably point-by-point/bullets)

Author Response

I wrote the answer in the attached file.

Reviewer 2 Report (Previous Reviewer 3)

The authors corrected many issues.

There are, however, some minor issues left in the paper.

Figure 6 has low quality. It seems to be bitmap format which is not a good choice for text based figures.

"Figure 7. The confusion matrix of the LSTM model with the best classification results."

What is the reason to include only for LSTM and not for GRU and RNN?

It would be interesting to see statistical results like min, avg, max, std accuracy, or at least include the number of executions, since the algorithms have stochastic behavior.

Author Response

I wrote the answer in the attached file.

Reviewer 3 Report (New Reviewer)

In this research, the authors suggest a method for detecting ADHD in kids.

They developed a gaming platform for the early detection and diagnosis of attention deficit hyperactivity disorder in kids, and collected skeletal data from those kids while they played the game with five Azure Kinect devices fitted with depth sensors.

The collected data was put into groups with the help of a bidirectional layer, a weighted cross-entropy loss function, and a set of algorithms called the Recurrent Neural Network (RNN) series.

I appreciate the general idea and the axis of application. However, I do have some suggestions that might help with the overall clarity of this paper:

- First, try to include a summary paragraph at the end of the introduction.

- Several paragraphs, including 3.2 and 3.4, are highlighted in red.

- Insert a few lines between 3 and 3.1

- Change the title of Section 4

- Fifth, your related work in this area needs extensive editing and expansion.

- If possible, you can enrich your introduction with some work in the medical field in general before talking about your specific axis (ADHD ).

- Below is some work for medical classification based on deep learning and machine learning, You can use the following to highlight your related work: 

Gasmi, K. (2022). Improving Bert-Based Model for Medical Text Classification with an Optimization Algorithm. In: Advances in Computational Collective Intelligence. ICCCI 2022. Communications in Computer and Information Science, vol 1653. Springer, Cham. https://doi.org/10.1007/978-3-031-16210-7_8

Gasmi, Karim. "Medical Text Classification based on an Optimized Machine Learning and External Semantic Resource." Journal of Circuits, Systems and Computers (2022).

- Compare the results obtained with another deep learning model other than LSTM. 

- I haven't been able to find a detailed discussion in the related work section, so I'd appreciate it if you could add a table or some other way to compare the proposed work. 

 - Seven, explain why GRU outperforms LSTM in several scenarios.

- Elucidate your method in greater detail.

- Why is there only one accuracy value listed in all the tables, but three for the other metrics?

- Tenth, refine Figure 6

Author Response

I wrote the answer in the attached file.

This manuscript is a resubmission of an earlier submission. The following is a list of the peer review reports and author responses from that submission.

Round 1

Reviewer 1 Report

1. The manuscript structure is moderate. It is well elaborated in their applied technology.
2. What is the novelty of the suggested approach?

3. Amount of literature work presented in the paper is less. It should be exhaustive and should be presented in very precise manner. The conclusions for the group of related papers should also be presented.

4. The work requires a comparative study with the latest existing method to justify the contribution and effectiveness. Justify.
5. A few latest studies may be incorporated. 

Reviewer 2 Report

This paper intends to analyze children's attention deficit hyperactivity disorder (ADHD). The authors use deep learning to automatically analyze skeleton data to identify such disorder in its early stages. Even though the problem is interesting, the work does not adequately describe and justify major aspects in various parts of the manuscript:

1) The motivation is weak as it is only argued that "there has been no study of ADHD classification using only skeleton data". My understanding is that a diagnosis of ADHD also takes into account inattentiveness and other factors outside of physical activity (e.g., difficulty concentrating and focusing). Authors, however, only paid attention to physical movement, leading one to believe that such a diagnosis type is insufficient and can produce misleading diagnosis results.

2) The related work describe four different articles. I believe that there are too many articles tackling the same problem that have not been considered in the analysis. In this sense, an in-depth analysis of related work is needed

3) The core content of the paper is located in Section 3, where the proposed method is described along with the data acquisition stage. Taking into account the information provided, it is not possible to understand the proposed method nor the data acquisition process. In particular, two physical movement types are considered (standby and game), but it not clear if they correspond to snapshots of a given time or a sequential of images captured at consecutive moments. Furthermore, the algorithm employed for the joints extraction is not detailed. In this regard, there are many different approaches for identifying the key points in the stickman, so that an appropriate justification of the selected approach is needed.

4) The ADHD diagnosis based on skeleton data should be compared to similar approaches that include other signs associated to ADHD.

Reviewer 3 Report

The article Deep-Learning-Based ADHD Classification Using Children’s Skeleton Data Acquired through the ADHD Screening Game presents the research results achieving good accuracy on predicting ADHD using only skeleton data for the machine learning algorithms.

The paper is basically good quality, there are however some minor problems which should be corrected.

For theseis reason” - typo

3.4

“The LSTM-based deep learning algorithm was used to classify the ADHD, ADHD-

RISK, and Normal groups.”

“The RNN models used were GRU, RNN, and LSTM.”

Not clear. Inconsistent sentences or different experiments?

Figure 6.

RNN vs LSTM - figure and description are inconsistent.

About the results: how many ML instances were executed for the different parameters?

Figure 7.

The figure and its font size are too big.
